# β-Alanine Metabolism Leads to Increased Extracellular pH during the Heterotrophic Ammonia Oxidation of *Pseudomonas putida* Y-9

**DOI:** 10.3390/microorganisms11020356

**Published:** 2023-01-31

**Authors:** Ming Nie, Kaili Li, Zhenlun Li

**Affiliations:** 1Chongqing Key Laboratory of Soil Multiscale Interfacial Process, College of Resources and Environment, Southwest University, Chongqing 400716, China; 2School of Chemical Engineering, University of Queensland, Brisbane 4072, Australia

**Keywords:** heterotrophic ammonia oxidation, pH, metabolites, metabolic pathway, β-alanine metabolism, RT-qPCR

## Abstract

The mechanisms underlying the increase in external pH caused by heterotrophic nitrification and aerobic denitrification microorganisms during ammonia oxidation were unclear. This work demonstrated that after culturing Pseudomonas putida Y-9 for 60 h in a medium with ammonium nitrogen as the sole nitrogen source at an initial pH of 7.20, the pH value increased to 9.21. GC-TOF-MS analysis was used to compare the significantly regulated metabolites and related metabolic pathways between different time points. The results showed that the consumption of H+ in the conversion of malonic acid to 3-hydroxypropionic acid in the β-alanine metabolic pathway was the main reason for the increase in pH. RT-qPCR confirmed that the functional gene *ydfG* dominated the consumption of H+. This study provides new research ideas for the change of external pH caused by bacterial metabolism and further expands the understanding of the interaction between bacteria and the environment.

## 1. Introduction

Microbial oxidation of ammonia to nitrite is the first step of the nitrification process, which is a critical part of the global biogeochemical nitrogen cycle. Under aerobic conditions, it is mainly driven by three aerobic autotrophic microorganisms, including ammonia-oxidizing bacteria (AOB), ammonia-oxidizing archaea (AOA), and comammox bacteria (complete oxidation of ammonia to nitrate) [1]. The ammonia oxidation process is an enzymatic process of microorganisms that converts ammonium nitrogen into nitrite nitrogen through oxidation. A series of oxidation reactions that occur during this process release H^+^ [2]. This indicates that ammonia oxidation may lead to an increase in acidity and a decrease in pH value [3].

Heterotrophic nitrification and aerobic denitrification microorganisms (HNADMs) can also heterotrophically perform ammonia oxidation. HNADMs have been extensively studied for their ability to simultaneously complete nitrification and denitrification under aerobic conditions [4]. HNADMs range from the aerobic denitrification species *Paracoccus pantotrophus*, which was reported as early as the 1980s, to species such as *Rhodococcus* sp., *Chryseobacterium* sp., *Pseudomonas* sp., *Bacillus cereus*, *Ochrobactrum* sp., and *Acinetobacter* sp. [5]. Contrary to the expectation that ammonia oxidation would lead to a decrease in external pH, recent studies on the nitrogen cycle driven by HNADMs have shown that HNADMs can lead to an increase in extracellular pH during heterotrophic ammonia oxidation. For example, in the process of removing ammonium nitrogen by *P. mendocina* X49, it was observed that the pH of the medium increased from about 6.80 to 8.28–8.64 [5]. Our previous studies focusing on the *Pseudomonas putida* Y-9-driven heterotrophic ammonia oxidation process illustrated the same phenomenon. Cultured with ammonia nitrogen as the only nitrogen source for 60 h, Y-9 causes the extracellular pH to rise continuously from 7.20 to 9.21. However, the reasons underlying this phenomenon remain unclear.

Available research attributes the phenomenon of bacteria causing an increase in the pH of the external environment to two main reasons: (a) One reason is the production of ammonium nitrogen or amines during bacterial metabolism. Ammonium nitrogen comes from the ammoniation and mineralization of organic nitrogen [6]. The production of amines is due to the action of amino acid decarboxylase [7]; (b) The other reason is the denitrification process of bacteria. The conversion of nitrite in the denitrification process into nitric oxide produces hydroxyl groups, which increase the external pH [8]. However, the above conclusions solely explain the autotrophic bacteria metabolism during nitrification, which cannot be used to interpret the increase in external pH for the heterotrophic ammonia oxidation bacteria when they are cultured in a neutral medium (pH = 7.19) with ammonium nitrogen as the sole nitrogen source. Several prior studies have shown that carbon sources are also a factor that leads to the change in extracellular pH [9]. However, the mechanism of the increase in extracellular pH caused by carbon sources is not clear.

Pseudomonas putida Y-9, a cold-tolerant strain of heterotrophic nitrification and aerobic denitrification [10], was used to study metabolic-induced changes in extracellular pH during heterotrophic ammonia oxidation. The nitrogen transformation and extracellular pH changes of strain Y-9 during its growth were investigated, and the mechanisms of changing the medium’s pH through the continuous changes of metabolites were explored. We found that strain Y-9 was able to cause a sustained increase in extracellular pH during a 60-h culture for ammonia oxidation with ammonium nitrogen as the sole nitrogen source, which proved that strain Y-9 regulated external pH by a mechanism other than nitrogen conversion. Further analysis inferred that nitrogen transformation was not the decisive factor for strain Y-9 to change the pH of the medium. The main reason is that the strain β-alanine metabolism consumes H^+^, thereby gradually increasing external pH. This study revealed another metabolic path leading to the change of bacterial extracellular pH, which offers new insights for studying the change of external environmental pH caused by bacterial metabolism.

## 2. Materials and Methods

### 2.1. Microorganism and Culture Media

The heterotrophic nitrification and aerobic denitrification bacterium *P. putida* Y-9 (KP410740) used in this study were previously isolated by Huang et al. [10]. The nitrification medium (NM) per liter comprised 1.3 g K_2_HPO_4_·3H_2_O, 0.24 g (NH_4_)_2_SO_4_, 2.56 g CH_3_COONa, 0.05 g FeSO_4_·7H_2_O, and 0.1 g MgSO_4_·7H_2_O.

The initial pH of all the media mentioned above was adjusted to 7.20 with NaOH. Conical flasks (250 mL capacity) containing 100 mL NM were autoclaved for 15 min at 115 °C. During incubation, samples were withdrawn and measured using a spectrophotometer. The optical density (OD_600_) of the bacterial cells was measured from the absorbance at 600 nm.

### 2.2. Effect of Different Carbon Sources on Extracellular pH and Growth of Strain Y-9

In the carbon source experiments, one of the three carbon sources (sodium acetate, glucose, or sodium citrate) was added to 100 mL of NM instead of CH_3_COONa. Cells killed by high temperature were inoculated into NM as a control. Approximately 20 mL of culture medium was taken out to determine pH and OD_600_ at set intervals.

### 2.3. Ammonium Oxidation Capacity of Y-9

In order to explore the influence of Y-9 on extracellular pH during the ammonia oxidation process, (NH_4_)_2_SO_4_ was used as the sole nitrogen source. A single colony of Y-9 was inoculated into 100 mL LB medium and cultured at 150 r/min and 15 °C for 36 h. Then, 8 mL of the precultured strain Y-9 were harvested by centrifuging at 4000 rpm for 8 min, and the pellet was washed twice with sterile water before inoculation into 100 mL of the medium containing different initial concentrations of ammonium. The cultures were incubated at 15 °C under aerobic conditions with shaking at 150 rpm. The medium without inoculation was used as a control group. Samples were periodically (12 h, 18 h, 24 h, 36 h, and 60 h) taken from the cultures to determine the optical density at 600 nm (OD_600_), pH, and concentrations of NH_4_^+^, NO_2_^−^, NO_3_^−^, NO, and total nitrogen (TN). All experiments were conducted in triplicate.

A total of 20 mL of culture medium was centrifuged at 8000 rpm for 5 min, and the supernatant was used to determine the pH as the extracellular pH. The cells collected after centrifugation were washed twice with sterile water and then broken with a cell disruptor. The cell-disrupted mixture was resuspended with 10 mM KCl to a volume of 20 mL to determine the pH value as the intracellular pH value.

### 2.4. Detection of Extracellular Metabolites Produced by P. putida Y-9

LB solid plates were used to activate *P. putida* Y-9, which was then enriched in a 100 mL LB broth medium and cultured in the shaker at 150 rpm, and 15 °C for 36 h. Then, 80 mL of precultured Y-9 strain was taken out and centrifuged for 5 min at 6000 rpm. Next, the pellet was washed twice with sterilized pure water. The supernatant was discarded, and the pellet was inoculated into a 100 mL basal medium with an initial pH of 7.20. The cultures were incubated at 15 °C under aerobic conditions with shaking at 150 rpm. After culturing for 12 h, 18 h, 24 h, 36 h, and 60 h, centrifugation was carried out at 8000 rpm for 5 min, and the supernatant was taken to determine the pH value and metabolites. The control test was carried out without inoculation and preculture. All experiments were repeated six times.

GC-TOF-MS analysis was performed using an Agilent 7890 (Agilent Technologies, Santa Clara, CA, USA) gas chromatograph coupled with a time-of-flight mass spectrometer. The system utilized a DB-5MS capillary column. A total of 1 μL aliquot of the sample was injected in the splitless mode. Helium was used as the carrier gas, the front inlet purge flow was 3 mL min^−1^, and the gas flow rate through the column was 1 mL min^−1^. The initial temperature was kept at 50 °C for 1 min, then raised to 310 °C at a rate of 10 °C min^−1^, and kept for 8 min at 310 °C. The injection, transfer line, and ion source temperatures were 280, 280, and 250 °C, respectively. The energy was −70 eV in electron impact mode. The mass spectrometry data were acquired in full-scan mode with the *m*/*z* range of 50–500 at a rate of 12.5 spectra per second after a solvent delay of 6.27 min. Raw data analysis, including peak extraction, baseline adjustment, deconvolution, alignment, and integration, was performed on Chroma TOF (V4.3x, LECO) software. LECO-Fiehn Rtx5 database was used for metabolite identification by matching the mass spectrum and retention index. Finally, the peaks detected in less than half of quality control (QC) samples or relative standard deviation (RSD) > 30% in QC samples were removed [11,12].

### 2.5. Real-Time Quantitative PCR

The expression levels of three functional genes associated with key metabolites were detected by real-time quantitative PCR (RT-qPCR). Total RNAs of the bacteria were extracted using the Trizol reagent (Life Technologies, Carlsbad, CA, USA). The cDNA was synthesized using PrimeScript™RT reagent Kit with gDNA Eraser (Takara, Japan). A ChamQ^TM^ Universal SYBR^®^ qPCR Master Mix (Vazyme, China) was used for qPCR sample detection. The qPCR instrument was a qTOWER3G (Analytikjena, Germany). The PCR conditions were as follows: 95 °C (initial activation temperature) for 3 min, then 95 °C (denaturation temperature) for 5 s, 60 °C (annealing temperature) for 5 s, and 72 °C (extension temperature) for 15 s by 40 cycles. All RT-qPCR experiments were repeated at least three times. All data were determined by the 2^−ΔΔCt^ method [13] and normalized to the 16S rRNA reference gene in each sample. The primer sequences for qPCR were as follows: *ydfG* (F: CGTGAAGGCATCACCGTCCATG; R: TGTCCAGACCTTCGACCACCTG), *rutE* (F: CGAACTGGGAGCGTTACCTGAAC; R: GCTGCAAAGGTGGTCTTGGAAATG), *PP-0596* (F: GCACCAGGAACGTCGTCGATATC; R: GCCACCGAAGCGTACATAGAAGC), 16S rRNA (F: GAACGCTAATACCGCATACGTCC; R: ATCATCCTCTCAGACCAGTTAC).

### 2.6. Analytical Methods

NO was detected using the nitrate reductase method, in accordance with the NO kit instruction designed by the Nanjing Jiancheng Bioengineering Institute, China [14]. TN was determined by measuring the absorbance value at 220 nm, subtracting two times the background absorbance value at 275 nm after the sample (without centrifugation) was digested using alkaline potassium persulfate. NH_4_^+^, NO_2_^−^, and NO_3_^−^ concentrations in the supernatant were measured after the samples were centrifuged at 8000 rpm for 5 min. NH_4_^+^, NO_2_^−^, and NO_3_^−^ concentrations were determined using the indophenols blue method, the hydrochloric acid photometry method, and the N-(1-naphthalene)-diaminoethane spectrophotometry method, respectively, in line with the guidelines set by the State Environmental Protection Administration of China (2012).

The enriched metabolic pathways and impact values of significantly different metabolites were analyzed with MetaboAnalyst 5.0 (https://www.metaboanalyst.ca, accessed on 16 June 2022).

### 2.7. Statistical Analysis and Graphical Work

Microsoft Excel 2010 and SPSS Statistics 19 were used to analyze the data. Hiplot (Scientific research data visualization platform) and Origin 2018 were employed in generating graphs. The results were presented as means ± SD (standard deviation of means).

## 3. Results

### 3.1. Effect of Different Carbon Sources on the Extracellular pH of P. putida Y-9

The effect of different carbon sources on the extracellular pH of Y-9 was evaluated. Sodium citrate, sodium acetate, or glucose were used as the sole carbon sources to culture Y-9 for 60 h, and the cultured dead cells were used as a control (Figure 1). When sodium citrate or sodium acetate was used as the carbon source, Y-9 led to a continuous increase in extracellular pH during the whole culture process, and the pH of the culture medium increased from 7.20 to 9.00 (sodium citrate) and 9.21 (sodium acetate), respectively. In the first 24 h of culture with glucose as the sole carbon source, Y-9 caused the extracellular pH to decrease from 7.20 to 5.52, but in the later stage of culture, the extracellular pH increased from 5.52 to 6.85. Given the extracellular pH change curve of different carbon sources, sodium acetate, which led to the highest increase in extracellular pH, was selected as the typical carbon source for further research.

### 3.2. P. putida Y-9 Upregulates External pH during the Heterotrophic Ammonia Oxidation

The ammonia oxidation process of strain Y-9 at an initial pH of 7.20 was evaluated, as well as the effect of Y-9 on the pH of the medium during cultivation (Figure 2). The results showed that the growth of strain Y-9 increased the extracellular pH from the initial condition of 7.20 to 9.21 after 60 h of culture. During the whole culture process, the intracellular pH (pHi) was stable at 7.00–7.20. The bacteria have a pH homeostasis mechanism, keeping the internal pH stable in the neutral range, which is consistent with the results of most bacterial internal pH studies [8]. During the nitrogen conversion process of Y-9, the total nitrogen dropped from the initial 49.76 to 41.44 mg/L in 24 h. After that, there was little change in total nitrogen, indicating that 8.32 mg/L of ammonium nitrogen was converted to gaseous nitrogen. After culturing strain Y-9 for 24 h, the presence of NO in the medium was detected, which indicated that Y-9 underwent ammonia oxidation during the culturing phase. This conclusion is consistent with our previous study that showed a new aerobic ammonia oxidation pathway by Y-9, which directly oxidizes ammonium nitrogen to N_2_O through NO under aerobic conditions without hydroxylamine as the intermediate [10]. Ammonium nitrogen was used as the sole nitrogen source of the culture medium, with an initial concentration of 48.80 mg/L. After 24 h of culture, ammonium nitrogen concentration was decreased to 1.25 mg/L. After 36 h of culture, ammonium nitrogen was completely consumed. The accumulation of NO_2_^−^ and NO_3_^−^ was not detected at any of the time points. OD_600_ reached a peak value of 1.247 at 36 h.

### 3.3. Analysis of Significantly Different Metabolites of Y-9 Strains at Different Culture Times

In order to investigate the medium pH change caused by Y-9 metabolites, GC-TOF-MS was used to study the nontarget metabolites with initial culture pH conditions of 7.20. After peak extraction and matching of the original data, 186 peaks were retained, and 63 known metabolites were measurable. Metabolites satisfying the criteria VIP > 1 and *p* < 0.05 were identified as differentially expressed metabolites; those with a fold change (FC) > 1 were considered upregulated, while those with an FC < 0.9 were considered downregulated. Four groups of metabolites were obtained as comparative data, including S1 (18 h vs. 12 h), S2 (24 h vs. 12 h), S3 (36 h vs. 12 h), and S4 (60 h vs. 12 h), and 19 metabolites were significantly regulated by KEGG annotation (Table 1). These metabolites included one from the group of alcohols and polyols, one of the sulfamic acid derivatives, eight carboxylic acids and derivatives, two hydroxy acids and derivatives, one fatty acyl, one imidazopyrimidine, one of keto acids and derivatives, and four unclassified compounds.

### 3.4. Analysis of Metabolic Pathway Enrichment of Y-9 Differential Metabolites

The differential metabolites were mapped to the Kyoto Encyclopedia of Genes and Genomes (KEGG) database (http://www.genome.jp/kegg/ accessed on 1 August 2022) for metabolic pathway construction [15]. Through comprehensive analysis (including enrichment and topological analyses) of the pathways where the differential metabolites were located, the pathways were further screened, and the key pathways with the highest correlation with metabolite differences were identified. As shown in Figure 3, each bubble in the bubble chart represents a metabolic pathway. Seven metabolic pathways showed significant enrichment (*p* < 0.05), and five metabolites were mapped to β-alanine metabolism (Figure 4) and glyoxylate and dicarboxylate metabolism (Figure 5). In terms of the impact value and significance of metabolites, these two metabolic pathways were relatively important, of which the largest impact value was 0.80 for β-alanine metabolism. Although five metabolites were mapped to glyoxylate and dicarboxylate metabolism, their impact value was as low as 0.08. Three metabolites were mapped to sulfur metabolism and propanoate metabolism, but their impact values were relatively low. Though only two metabolites mapped between the Citrate cycle (TCA cycle) and Pantothenate and CoA biosynthesis, their influence value ranked second after β-alanine metabolism. Two metabolites were mapped in monobactam biosynthesis with an impact value of 0.

### 3.5. Correlation Analysis between FC of Metabolite and FC of the Medium’s pH

From the analysis of substances with significant regulation in all comparative metabolite data groups, we found that in addition to oxalic acid, the FC of the upregulated substances continued to rise in the S1, S2, and S3 groups, whereas the FC of the downregulated substances continued to decline in the S1, S2, and S3 groups, and rose slightly in the S4 group. This phenomenon was consistent with the accelerated rise of pH in the early stage of culturing and the slow rise of pH from 36 h to 60 h. Therefore, a Pearson correlation analysis was conducted between the rising FC of the medium’s pH and the significantly regulated FC of the metabolites (Figure 6). From the correlation heat map, it is clear that there was a significant negative correlation between the medium’s pH and the downregulated metabolite malonic acid 1 but no significant correlation with maleamate 4. The medium’s pH positively correlated with the upregulated metabolites 3-hydroxypropionic acid 1, alanine 1, serine 1, and cyclohexylsulfamic acid 1, but there was no significant correlation with myo-inositol, oxalic acid, and pantothenic acid. The correlation analysis showed that the increase in extracellular pH was significantly related to β-alanine metabolism.

### 3.6. RT-qPCR Validation of Related Genes in β-alanine Metabolism

Genes *ydfG* and *rutE* associated with the metabolism of malonate to 3-HP were identified from the whole-genome sequencing data of Y-9. *ydfG* encodes 3-hydroxypropionate dehydrogenase or malonate semialdehyde reductase, which can reduce malonate semialdehyde to 3-HP (Figure 7). *rutE* encodes malonate semialdehyde reductase, and previous studies have shown that its function is consistent with *ydfG* [16]. The enzyme encoded by gene *PP-0596* is responsible for transforming malonate semialdehyde into β-alanine. RT-qPCR verification on these three genes was performed using the 12-h expression as the reference. Figure 8 shows the expression of these three genes treated at 18 h, 24 h, 36 h, and 60 h. After 18 h, all three genes were upregulated, but FC was less than 1. *ydfG* was upregulated by a 2.65 ± 0.27 fold and a 0.46 ± 0.06 fold at 24 h and 36 h, respectively, while *rutE* and *PP-0596* were downregulated. After 60 h, all three genes were downregulated.

## 4. Discussion

### 4.1. Response of Extracellular pH of Y-9 to Different Carbon Sources

Carbon source is one of the important factors which affect the extracellular pH. Previous studies have shown that the extracellular pH of Candida albicans ranged from 4.0 to 7.0 as a carbon source with acetate within 24 h, and a decrease of extracellular pH (ranging from 4.0 to 2.5) was observed when glucose was used as the carbon source [17]. Sánchez Clemente et al. [9] cultured three strains (*Escherichia coli* ATCC 25922, *Pseudomonas putida* KT2440, and *Pseudomonas pseudoalcaligenes* CECT 5344) with different carbon sources and found that alkalinization of the medium when citrate as the carbon source, while a slightly acidic of the medium when glucose was added as the carbon source. In our study, both increases in extracellular pH of Y-9 were observed when acetate and citrate were used as carbon sources individually, which were in accord with previous studies [9]. Interestingly, we also found that the extracellular pH showed a trend of decreasing first and then rising when glucose was added as the carbon source, which differed from the findings presented. These relationships may partly be explained by the fact that Y-9 has the ability to regulate extracellular pH. In this experiment, sodium acetate caused a continuous increase in extracellular pH. Therefore, sodium acetate was used as a carbon source for further research.

### 4.2. Nitrogen Metabolism Is Not the Reason Why Y-9 Causes the Increase in Extracellular pH

According to previous studies, the increase in external pH value caused by bacteria was attributed to the accumulation of ammonium nitrogen or denitrification [18]. Zhang et al. reported that in the process of using bacteria to remove Pb^2+^, the strain increased the pH of wastewater from four to eight within 7 days, and the rising trend of pH correlated with the accumulation of ammonium nitrogen [19]. The nitrification process can lead to the opposite result. For instance, the ammonia oxidation process carried out by AOB can lead to a drop in pH [20]. However, our research results contradict those from the previous studies. Cultivating Y-9 with ammonium nitrogen as the sole nitrogen source for ammonia oxidation resulted in a continuous increase in the pH of the medium. He et al. also found a similar phenomenon when studying the nitrification process of *Arthrobacter arilaitensis* Y-10 [21]. In their nitrification test, low-concentration ammonium nitrogen was used as the only nitrogen source. After culturing strain Y-10 for 24 h, the ammonium nitrogen concentration rapidly decreased from the initial 9.62 to 0 mg/L, and the pH value gradually increased from 7.30 to 8.31. Yang et al. studied the heterotrophic nitrification ability of *Acinetobacter* sp. JR1 with ammonium as the only nitrogen source at an initial pH of 4.50. Ammonium nitrogen was completely removed within 24 h, and the pH of the medium increased continuously from 4.50 to 9.50 within 48 h. During the whole ammonium removal process, no accumulation of nitrate and nitrite was detected [22]. In this study, assimilation and ammonia oxidation was performed during the culturing phase of Y-9. Ammonium nitrogen was completely exhausted after 36 h of culture, and the pH of the medium continued to rise to 60 h, indicating that the rise in pH of the medium was not entirely caused by the nitrogen transformation path.

### 4.3. Analysis of the Reasons for the Increase in Extracellular pH Using Metabolites

To further study why Y-9 causes the increase in extracellular pH, we measured the metabolites of Y-9 at different culture time points. Since the pH of the medium gradually increased with time, the study focused on the substances that were upregulated or downregulated at each time point compared with the metabolites of 12 h. The substances that were upregulated in all control groups were myo-inositol, cyclohexylsulfamic acid 1, serine 1, 3-hydroxypropionic acid 1, alanine 1, oxalic acid, and pantothenic acid. The substances that were downregulated are malonic acid 1 and maleamate 4. In addition to myo-inositol, these metabolites are all organic acids, indicating that the significantly regulated metabolites do not directly increase the medium’s pH. We speculate that this phenomenon may be caused by the consumption of H^+^ or the release of OH^−^ in the medium during the metabolism of Y-9. In the KEGG’s description of metabolites, the production of serine 1 requires the consumption of H^+^ in the reaction of some metabolic pathways, such as KEGG reaction: R10986 (Aminoacetaldehyde + CO_2_ + NADH + H^+^ → L-serine + NAD^+^). In addition, 3-hydroxypropionic acid 1 can be generated from KEGG reaction: R01608 (3-oxopropanoate + NADH + H^+^ → 3-hydroxypropanoate + NAD^+^), and 1 H^+^ is consumed in the reaction process. Moreover, the production of oxalic acid is accompanied by the production of H_2_O_2_, such as the KEGG reaction: R00466 (Glyoxylate + Oxygen + H_2_O → Oxalate + Hydrogen peroxide). Fe^2+^ was added to the culture medium in this work, and the Fenton reaction between Fe^2+^ and H_2_O_2_ released OH^−^ [23]. However, the same metabolites appear in a variety of different metabolic pathways. For example, 3-hydroxypropionic acid (3-HP) is recognized by the U.S. Department of Energy as one of the most promising value-added chemicals that can be obtained from biomass through the CoA-independent pathway, malonyl-CoA pathway, and β-alanine pathway [24]. To further explore whether the related reactions of H^+^ consumption or OH^−^ release did occur in these metabolites, metabolic pathway enrichment analysis was performed on the above 19 metabolites.

### 4.4. Effect of β-alanine Metabolism and Glyoxylate and Dicarboxylate Metabolism on the Rise of pH in the Culture Medium

After comprehensively analyzing the significance of the enriched metabolic pathways, the number of metabolites mapped, and the impact value, it was deduced that the metabolic pathways most likely to affect the increase of the medium’s pH were β-alanine metabolism and glyoxylate and dicarboxylate metabolism. From the comprehensive analysis of the enrichment and metabolic pathway of metabolites, β-alanine metabolism was the metabolic pathway with the greatest effect on the medium’s pH. Five metabolites were labeled in β-alanine metabolic pathway map, as shown in Figure 4. Namely, the metabolites mapped on this metabolic pathway showed significant upregulation (3-hydroxypropionic acid and pantothenic acid) and significant downregulation (malonic acid, 4-aminobutyric acid, and β-alanine). From the comparative data and metabolic routes of the four groups of metabolites, it was noticeable that there may be a transformation relationship between the continuously upregulated 3-HP and the continuously downregulated malonic acid. The two functional groups (carboxyl and β-hydroxyl) contribute to the versatility of 3-HP, which has a great potential for further transformation into useful chemicals. The first part of the pathway, from glucose to two acetyl-CoA, is achieved through endogenous sugar catabolism, comprising glycolysis. Malonyl-CoA is then obtained from acetyl-CoA using an acetyl-CoA decarboxylase. The last two steps, from malonyl-CoA to 3-HP, are catalyzed by malonyl-CoA reductase (MCR). The MCR has two distinct functional domains, where the first one is C-terminal and catalyzes malonyl-CoA’s reduction into malonate semialdehyde (MSA), while the second one is N-terminal and catalyzes MSA’s reduction into 3-HP [24]. Malonic acid can also be directly used as a substrate to produce 3-HP through the malonyl-CoA pathway [25]. There have been no reports of wild strains producing 3-HP with malonic acid as the substrate. In any case, there is H^+^ consumption in the production of 3-HP by the malonyl-CoA pathway, which is considered to be one of the reasons for the rise of the medium’s pH. Hydrogen ions are consumed through paths (1) and (2). The KEGG reaction is as follows:Acetyl-CoA + CO_2_ + NAD(P)H + H^+^ → Malonate semialdehyde + CoA + NAD(P)^+^(R00705/R00706) (1)
Malonate semialdehyde + NADH + H^+^ →3-Hydroxypropanoate + NAD^+^ (R01608)(2)

Five metabolites of glyoxylic acid and dicarboxylic acid metabolism were also mapped, including serine 1, oxalic acid, succinic acid, glycolic acid, and citric acid. As suggested in Figure 5, glycolic acid may also be transformed into oxalic acid, and hydrogen peroxide is produced in this transformation process. Hydrogen peroxide is produced through paths (3) and (4). The KEGG reaction process is as follows: Glycolate + Oxygen → Glyoxylate + H_2_O_2_ (R00475)(3)
 Glyoxylate + H_2_O + O_2_ → Oxalate + H_2_O_2_ (R00466)(4)

Since Fe^2+^ was added to the medium, the reaction of Fe^2+^ with H_2_O_2_ released OH^−^(H_2_O_2_ + Fe^2+^ → ·OH^−^ + OH^−^ + Fe^3+^), and it was concluded that this might also be one of the reasons for the rise of the medium’s pH. Gomez Toribio et al. reported that white-rot fungi induced the production of extracellular hydroxyl radicals through the quinone redox cycle, in which quinone redox produced H_2_O_2,_ and the Fenton reaction occurred [23].

In addition, with acetate as the carbon source, 3-HP can be synthesized through the malonyl-CoA pathway. Acetate can be converted into acetyl-CoA in glyoxylic acid and dicarboxylic acid metabolism, and then 3-HP is synthesized through the malonyl CoA pathway. The reaction from malonyl CoA to 3-HP is a part of the 3-hydroxypropionate and 3-hydroxypropionate/4-hydroxybutyrate cycle, which are two of the six pathways of autotrophic carbon dioxide fixation [26]. This indicates that the increase in pH of the medium may be related to the pathway of carbon sequestration.

### 4.5. Correlation Analysis and RT-qPCR Verification Support That the Rise in Extracellular pH Is Caused by β-alanine Metabolism

Combined with the previous metabolic pathway analysis, we demonstrated that 3-hydroxypropionic acid 1, pantothenic acid, and malonic acid 1 were mapped in β-alanine metabolism. We found that 3-hydroxypropionic acid 1 was significantly positively correlated with pantothenic acid and negatively correlated with malonic acid 1. This proves that 3-hydroxypropionic acid 1 and malonic acid 1 are converted through the pathway of β-alanine metabolism, and the increase of the medium’s pH significantly negatively correlated with malonic acid 1 and significantly positively correlated with 3-hydroxypropionic acid 1. Hence, the rise in the medium’s pH is mainly caused by the consumption of H in the metabolic process of malonyl-CoA to 3-hydroxypropionic acid 1. In addition, there was no significant correlation between the rise of the medium’s pH and oxalic acid, indicating that the Fenton reaction caused by glyoxylate and dicarboxylate metabolism was not the main factor leading to the change in the medium’s pH.

To prove that the continuous increase in the pH of the medium was mainly due to the consumption of H^+^ during the metabolism of malonic acid to 3-HP, we further performed RT-qPCR verification. Interestingly, from the results, the genes related to the metabolism of malonate to malonate semialdehyde were not found, so we speculated that Y-9 has distinct genes to accomplish this task. Gene *PP-0596* related to the metabolism of malonate semialdehyde to β-alanine was found in the analysis (Figure 7). RT-qPCR verification on these three genes was performed using the 12-h expression as the reference. Figure 8 shows the expression of these three genes treated at 18 h, 24 h, 36 h, and 60 h. After 18 h, all three genes were upregulated, but FC was less than 1, indicating that the upregulation of genes was due to the rapid growth of the strain. *ydfG* was upregulated at 24 h and 36 h, indicating that *ydfG* dominated the reduction of malonate semialdehyde to 3-HP. Although *rutE* was reported to have the same function as *ydfG*, its expression was downregulated at 24 h and 36 h, indicating that *rutE* was not involved in the reduction of malonate semialdehyde to 3-HP. The gene *PP-0596* was downregulated in both 24-h and 36-h treatments, which also indicated that malonate semialdehyde was not converted to β-alanine. All three genes were downregulated after the 60 h treatment, which may indicate that the growth of the strain had reached the plateau stage. From the coherent analysis, it can be concluded that *ydfG* dominates the reduction of malonate semialdehyde to 3-HP, which consumes H^+^ and leads to an increase in the medium’s pH.

## 5. Conclusions

When P. putida Y-9 was cultured with ammonia nitrogen as the sole nitrogen source, its metabolites led to a continuous increase in extracellular pH during the 60 h culture. The process of ammonia oxidation could not dominate the change of extracellular pH, and the most influential metabolic pathway was β-alanine metabolism. The persistent increase in the pH of the medium was mainly due to the consumption of H+ during the metabolism of malonic acid to 3-hydroxypropionic acid. RT-qPCR validation showed that the gene ydfG was involved in this process. The final change in extracellular pH was likely caused by a combination of factors, and further studies are needed to clarify whether there are other factors that affect the change in pH.

## Figures and Tables

**Figure 1 microorganisms-11-00356-f001:**
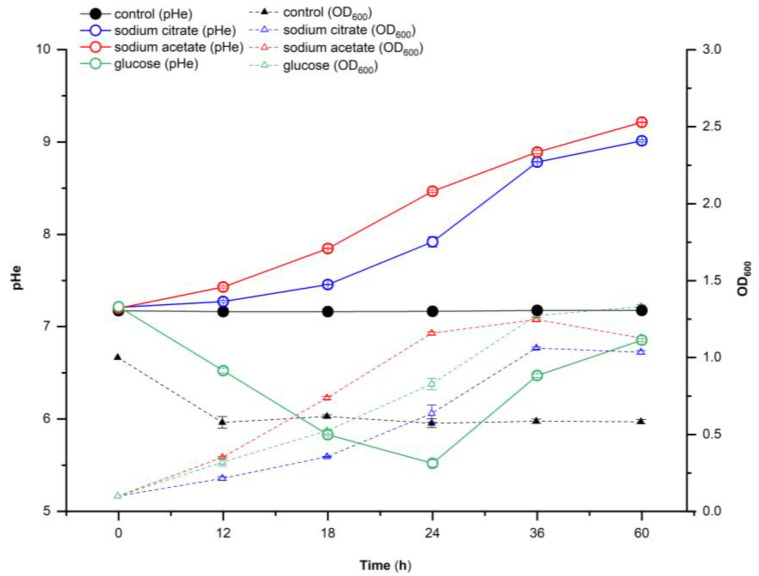
Effects of different carbon sources on the growth and extracellular pH of Y-9 at different growth times.

**Figure 2 microorganisms-11-00356-f002:**
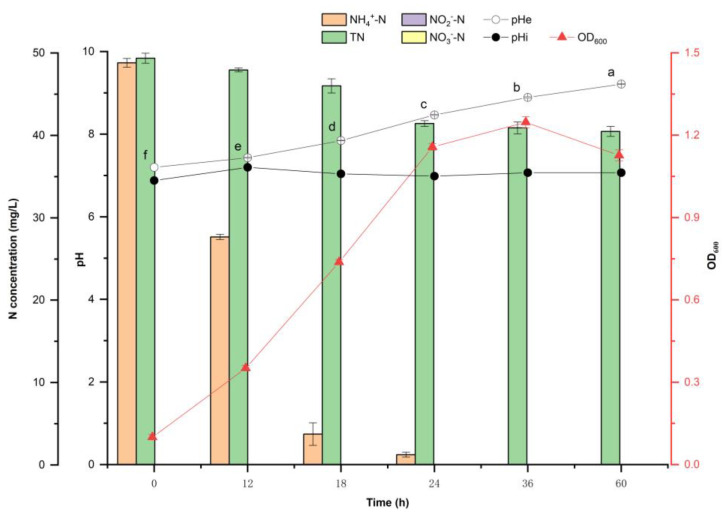
Nitrogen transformation characteristics and intracellular and extracellular pH changes of Y-9 at different growth times. Different lowercase letters indicate the significant difference of pHe at different growth times (*p* < 0.05).

**Figure 3 microorganisms-11-00356-f003:**
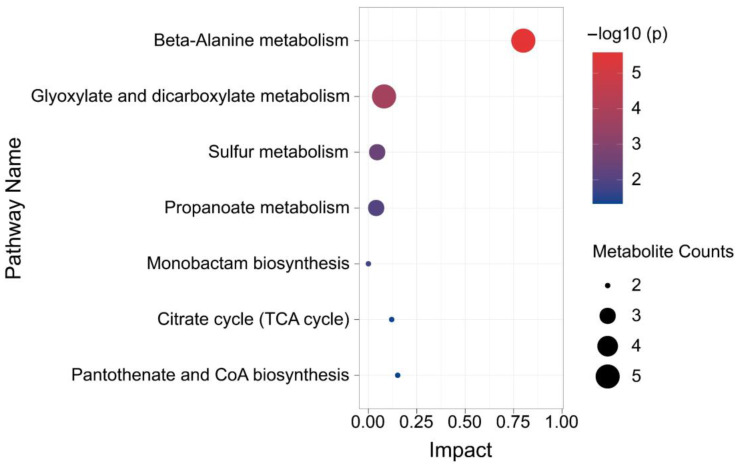
Metabolic pathway enrichment analysis bubble plot.

**Figure 4 microorganisms-11-00356-f004:**
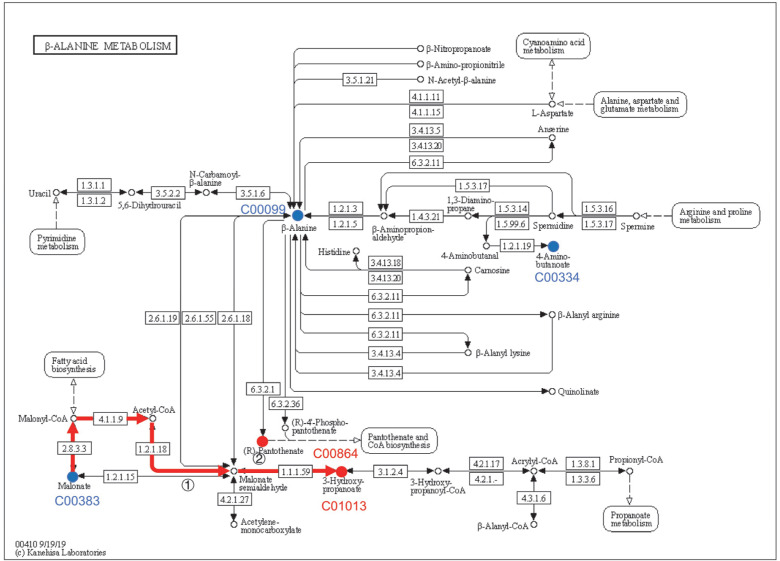
β-Alanine metabolism and related metabolites (red means upregulation, blue means downregulation).

**Figure 5 microorganisms-11-00356-f005:**
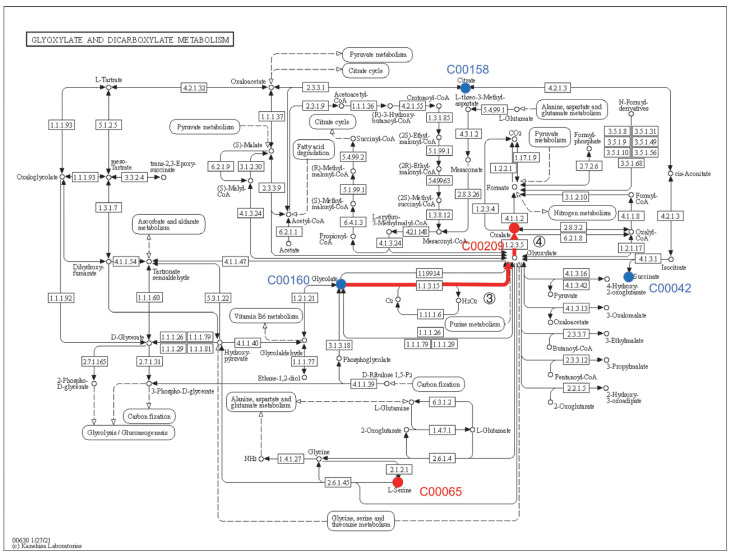
Glyoxylate and dicarboxylate metabolism and related metabolites (red means upregulation, blue means downregulation).

**Figure 6 microorganisms-11-00356-f006:**
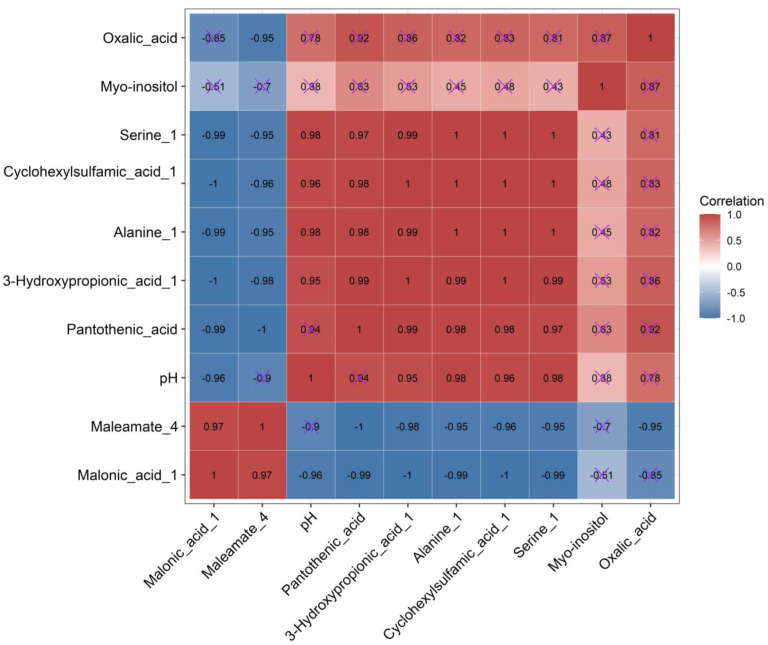
Heat map of the correlation between pH and metabolite fold change. × indicates that there is no significant correlation between the two metabolites.

**Figure 7 microorganisms-11-00356-f007:**
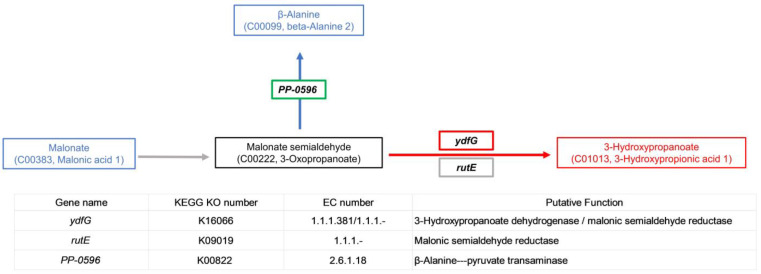
Metabolic pathways of key metabolites and description of related functional genes.

**Figure 8 microorganisms-11-00356-f008:**
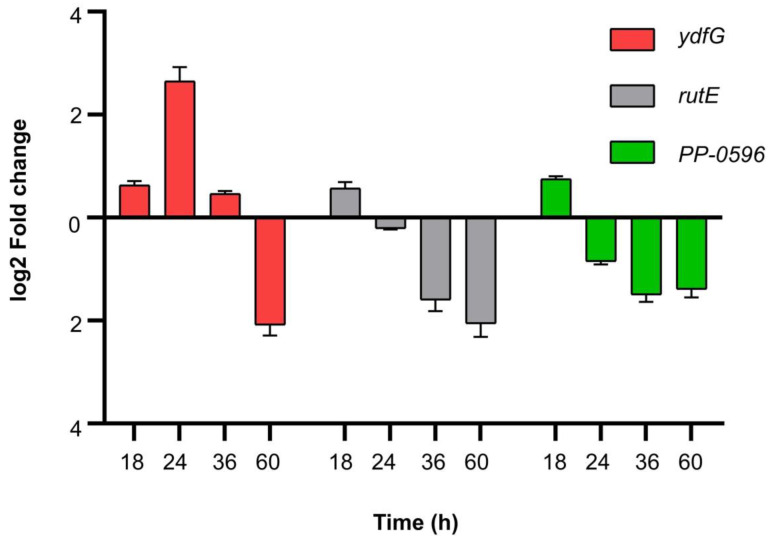
RT-qPCR analysis of related functional genes with different culture times.

**Table 1 microorganisms-11-00356-t001:** Fold change and classification of significantly different metabolites.

Metabolite	KEGG ID	Class	Fold Change			
			S1 (18 h VS. 12 h)	S2 (24 h VS. 12 h)	S3 (36 h VS. 12 h)	S4 (60 h VS. 12 h)
Myo-inositol	C00137	Alcohols and polyols	417,387.89	1,550,810.60	944,079.06	917,126.45
Syclohexylsulfamic acid 1	C02824	Sulfamic acid derivatives	111,839.17	1,661,137.85	2,703,052.95	2,550,359.01
Serine 1	C00065	Carboxylic acids and derivatives	2.57	18.58	30.65	30.59
3-Hydroxypropionic acid 1	C01013	Hydroxy acids and derivatives	2.18	11.01	15.71	14.65
Alanine 1	C01401	-	1.98	6.87	10.16	10.43
Oxalic acid	C00209	Carboxylic acids and derivatives	1.74	5.56	4.75	5.06
Pantothenic acid	C00864	Carboxylic acids and derivatives	0.83	498,303.28	634,225.94	628,839.31
Myristic acid	C06424	Fatty acyls	-	8.73	24.21	22.64
Sulfuric acid	C00059	-	-	1.51	2.36	2.36
Phenylphosphoric acid	C02734	-	-	1.43	-	-
Adenine	C00147	Imidazopyrimidines	-	-	-	1.52
2-Ketobutyric acid 1	C00109	Keto acids and derivatives	0.88	-	-	-
4-Aminobutyric acid 3	C00334	Carboxylic acids and derivatives	0.83	-	-	-
Beta-alanine 2	C00099	Carboxylic acids and derivatives	0.83	-	-	-
Malonic acid 1	C00383	Carboxylic acids and derivatives	0.78	0.53	0.38	0.41
Maleamate 4	C01596	-	0.68	0.10	0.00	0.04
Succinic acid	C00042	Carboxylic acids and derivatives	-	0.69	0.45	0.76
Glycolic acid	C00160	Hydroxy acids and derivatives	-	-	0.68	0.81
Citric acid	C00158	Carboxylic acids and derivatives	-	-	0.00	0.00

The fold change >1 means upregulated, and the fold change <1 means downregulated.

## Data Availability

Not applicable.

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
