# Peer review of "β-Alanine Metabolism Leads to Increased Extracellular pH during the Heterotrophic Ammonia Oxidation of *Pseudomonas putida* Y-9"

_microorganisms, 2023, doi:10.3390/microorganisms11020356_

Round 1

Reviewer 1 Report (Previous Reviewer 1)

Nie et al studied β-Alanine metabolism leading to increased extracellular pH during the heterotrophic ammonia oxidation of Pseudomonas putida Y-9. The authors performed extensive characterization of the putida cells under different energy sources and observed a mechanism of pH modulation by Y-9, which likely has roots in cellular metabolism. A detailed metabolites search was performed on one culture condition with the highest pH alteration (sodium acetate), shedding lights on how carboxylic acids and derivatives changed during cell culture. There are a few suggestions to improve this manuscript. First, when glucose was used as the carbon source, culture pH did not return to starting pH after 60 hours. Please extend culturing time and check if Y-9 could change the medium pH to alkaline pH. Second, clearly the metabolomics between 12-18 hours and 36-60 hours on both glucose and acetate culture conditions are worth investigation.

Author Response

Response to Review Comments

Manuscript ID: microorganisms- 2195775

Title: Heterotrophic oxidation during culturing of Pseudomonas putida Y-9 leads to increased extracellular pH due to β-alanine metabolism

We thank reviews for the comments, which are very helpful and have improved the overall quality of the paper. Below is a summary of our response to the review comments. Detailed modifications have been highlighted with red color in the revised manuscript.

Here is the reply to the queries.

Response to comments from Reviewer #1

[Comment] Nie et al studied β-Alanine metabolism leading to increased extracellular pH during the heterotrophic ammonia oxidation of Pseudomonas putida Y-9. The authors performed extensive characterization of the putida cells under different energy sources and observed a mechanism of pH modulation by Y-9, which likely has roots in cellular metabolism. A detailed metabolites search was performed on one culture condition with the highest pH alteration (sodium acetate), shedding lights on how carboxylic acids and derivatives changed during cell culture. There are a few suggestions to improve this manuscript. First, when glucose was used as the carbon source, culture pH did not return to starting pH after 60 hours. Please extend culturing time and check if Y-9 could change the medium pH to alkaline pH. Second, clearly the metabolomics between 12-18 hours and 36-60 hours on both glucose and acetate culture conditions are worth investigation.

[Response]

Very grateful to the reviewers' comments, which is helpful for improving our manuscript. In the experiment, the extracellular pH of strain Y-9 had no obvious change from 60 h to 72 h when glucose was as the sole carbon source. At the same time, OD600 had no obvious growth. For the above reasons, we did not display the 72-hour data in the manuscript.

Just as you say, the metabolomics study between glucose and sodium acetate is very fascinating, which can help further explore the reason for extracellular pH changes during different carbon sources conditions, and this is also the future research direction of our laboratory. In this paper, we mainly focus on the reason why aerobic denitrification microorganismY-9 causes the continuous increase in extracellular pH during the heterotrophic ammonia oxidation. Owing to the complexity of the metabolomics under different carbon sources, the sodium acetate was screened from different carbons sources as the typical carbon source for further research. We really appreciate your suggestion which is very valuable. We will delve deeply into the effects of Y-9 metabolites on extracellular pH under different carbon source conditions in our future studies.

We have checked all of the problems, and modified all errors.

Thanks for your reconsideration of our manuscript.

Yours sincerely,

Zhenlun Li

Reviewer 2 Report (Previous Reviewer 2)

The modifications have increased the quality of the manuscript.

Author Response

Response to Review Comments

Manuscript ID: microorganisms- 2195775

Title: Heterotrophic oxidation during culturing of Pseudomonas putida Y-9 leads to increased extracellular pH due to β-alanine metabolism

Response to comments from Reviewer #2

[Comment] The modifications have increased the quality of the manuscript.

[Response]Thank you for your affirmation of the results we have done.

Yours sincerely,

Zhenlun Li

Reviewer 3 Report (Previous Reviewer 3)

The paper is fine

Author Response

对审核意见的回应

手稿ID:微生物-2195775

题目:恶臭假单胞菌Y-9培养过程中的异养氧化导致β-丙氨酸代谢导致细胞外pH值升高

对审稿人评论的回应 #3

[评论]论文很好

[回应]感谢您对我们所取得的成果的肯定。

你的真诚,

李振伦

This manuscript is a resubmission of an earlier submission. The following is a list of the peer review reports and author responses from that submission.

Round 1

Reviewer 1 Report

In this manuscript, Nie and coworkers study the phenomenon of increasing external pH caused by HNADM bacteria Pseudomonas putida Y-9. Using growth assays, GC-TOF-MS analysis and RT-qPCR, the authors suggest that the consumption of H+ in the conversion of malonic acid to 3-hydroxypropionic acid in the β-alanine metabolic pathway was the main reason for the increase in pH.

I have several concerns that preclude publication of this manuscript in the current form:

1) Several prior studies have demonstrated that the carbon source, instead of the nitrogen metabolism, is the determining factor for culture pH variation. For example, when the type of carbon source is pyruvate, citric acid or acetic acid, a higher pH value will be produced after metabolism. When sucrose and glucose are used as the carbon source, the pH was decreased after microbial metabolism (Yang et al., 2011; Liu et al., 2015; Rubén Sánchez-Clemente et al., 2018; Rubén Sánchez-Clemente et al., 2020). In this study, sodium acetate was used as the sole carbon source, which likely explained the increase in pH during cell culture.

2) The GC-MS data suggested metabolic changes in several pathways, which are more likely a result of the altered growth phases (and/or altered medium pH) during culture vs. the reasons for pH change as proposed by the authors.

3)  The results are poorly presented for readers. For examples: a) missing citation (line 78); b) what does TN stand for? (Line 148 and Figure 1); c) “different letters indicate significant”? (Line 187); d) what is the meaning of “x” in Figure 5?

4) Since extracellular metabolites were screened, controls for cell death are needed.

Author Response

Response to Review Comments

Manuscript ID: microorganisms- 2114710

Title: Heterotrophic oxidation during culturing of Pseudomonas putida Y-9 leads to increased extracellular pH due to β-alanine metabolism

We thank reviews for the comments, which are very helpful and have improved the overall quality of the paper. Below is a summary of our response to the review comments. Detailed modifications have been highlighted with red color in the revised manuscript.

Here is the reply to the queries.

Response to comments from Reviewer #1

[Comment 1] Several prior studies have demonstrated that the carbon source, instead of the nitrogen metabolism, is the determining factor for culture pH variation. For example, when the type of carbon source is pyruvate, citric acid or acetic acid, a higher pH value will be produced after metabolism. When sucrose and glucose are used as the carbon source, the pH was decreased after microbial metabolism (Yang et al., 2011; Liu et al., 2015; Rubén Sánchez-Clemente et al., 2018; Rubén Sánchez-Clemente et al., 2020). In this study, sodium acetate was used as the sole carbon source, which likely explained the increase in pH during cell culture.

[Response] Several previous studies mainly focused on the effect of different carbon sources on extracellular pH. In this paper, sodium acetate was used as the sole carbon source to determine the metabolites at different growth times, which provided a new explanation for bacterial metabolism to change extracellular pH.

[Comment 2] The GC-MS data suggested metabolic changes in several pathways, which are more likely a result of the altered growth phases (and/or altered medium pH) during culture vs. the reasons for pH change as proposed by the authors.

[Response] We have conducted a series of correlation analyses and found that there is a significant correlation between metabolites of β-alanine metabolism and extracellular pH value, besides, this metabolic process consumes H+. In Y-9 growth stage, a number of metabolites are indeed produced, which have no significant correlation with extracellular pH value except those related to β-alanine metabolism.

[Comment 3] The results are poorly presented for readers. For examples: a) missing citation (line 78); b) what does TN stand for? (Line 148 and Figure 1); c) “different letters indicate significant”? (Line 187); d) what is the meaning of “x” in Figure 5?

[Response] a) The revised article has been added citations. b) TN stands for total nitrogen, and the supplementary description has been added in line 96. c) Different lowercase letters indicate the significant difference of pHe at different growth times (P<0.05). Annotations have been added to the revised article d) “x” indicates that there is no significant correlation between the two metabolites. Annotations have been added to the revised article.

[Comment 4] Since extracellular metabolites were screened, controls for cell death are needed.

[Response] Thank you for your valuable suggestions. In our study, the metabolites significantly related to extracellular pH changes continuously increased (3-Hydroxypropionic acid 1) or decreased (Malonic acid 1) throughout the culture period, so we believe that these metabolites come from the growth process of bacteria. In future studies, the control of cell death will be taken into account.

We have checked all of the problems, and modified all errors.

Thanks for your reconsideration of our manuscript.

Yours sincerely,

Zhenlun Li

Reviewer 2 Report

Generalities

The manuscript entitled “Heterotrophic oxidation during culturing of Pseudomonas putida Y-9 leads to increased extracellular pH due to β-alanine metabolism” aims to provide a metabolic explanation to the increase of pH during heterotrophic nitrification. I read the manuscript with very much interest. I only have few comments.

Specifications

1)      Line 96: The authors analysed ammonium, nitrite, nitrate, nitric oxide and total nitrogen. I wonder why nitrous oxide was not analysed. It is more stable than nitric oxide and it is routinely analysed in nitrificaction-denitrification studies,

2)      Line 139. Some characters cannot be seen.

3)      Figure 1 shows that 50 mg/L ammonium were removed in ca. 24h without nitrite nor nitrate accumulation. However, only 10 mg/L TN were removed. Which were the ammonium removal end-products?

4)      Table 1. Some words are cut. Please revise.

5)      Figure 5. It is not clear why some values are strikethroughed.

Author Response

Title: Heterotrophic oxidation during culturing of Pseudomonas putida Y-9 leads to increased extracellular pH due to β-alanine metabolism

We thank reviews for the comments, which are very helpful and have improved the overall quality of the paper. Below is a summary of our response to the review comments. Detailed modifications have been highlighted with red color in the revised manuscript.

Here is the reply to the queries.

Response to comments from Reviewer #2

[Comment 1] Line 96: The authors analysed ammonium, nitrite, nitrate, nitric oxide and total nitrogen. I wonder why nitrous oxide was not analysed. It is more stable than nitric oxide and it is routinely analysed in nitrificaction-denitrification studies.

[Response] Previous studies have proved that the only nitrogen source of Y-9 is ammonium nitrogen, and the end-product is N2O, and the previous research article[9] was cited for explanation.

[Comment 2] Line 139. Some characters cannot be seen.

[Response] Maybe character 2-ΔΔCt has caused your misunderstanding, Δ means Delta. It has been revised in the article.

[Comment 3] Figure 1 shows that 50 mg/L ammonium were removed in ca. 24h without nitrite nor nitrate accumulation. However, only 10 mg/L TN were removed. Which were the ammonium removal end-products?

[Response]Previous studies on Y-9 show that the end-products of Y-9 ammoxidation is N2O[9]. The removal of ammonium could be divided into two parts, one part of ammonium is converted to N2O (10 mg/L TN) and the other part of ammonium is assimilated.

[Comment 4] Table 1. Some words are cut. Please revise.

[Response] It has been revised in the article.

[Comment 5] Figure 5. It is not clear why some values are strikethroughed.

[Response]× indicates that there is no significant correlation between the two metabolites. Annotations have been added to the revised article.

We have checked all of the problems, and modified all errors.

Thanks for your reconsideration of our manuscript.

Yours sincerely,

Zhenlun Li

Reviewer 3 Report

the paper is fine for me

Author Response

Response to Review Comments

Manuscript ID: microorganisms- 2114710

Title: Heterotrophic oxidation during culturing of Pseudomonas putida Y-9 leads to increased extracellular pH due to β-alanine metabolism

Response to comments from Reviewer #3

[Comment]the paper is fine for me

[Response]Thank you for your affirmation of the results we have done.

Yours sincerely,

Zhenlun Li

Round 2

Reviewer 1 Report

Thank authors for the reply.The reviewer still believes that critical control experiments are needed for this report to be useful for scientific community. For example, in addition to the control of cell death, authors should explain why they use sodium acetate as carbon source and collect more data with sucrose or glucose as carbon source.

There are typos in the Keywords section. Eg. RT-qPCR 6 should be just "RT-qPCR".